# Evaluating Robustness to Dataset Shift
# via Parametric Robustness Sets

**Nikolaj Thams**[*]
Dept. of Mathematical Sciences
University of Copenhagen
Copenhagen, Denmark
thams@math.ku.dk

**Michael Oberst**[*]
CSAIL & IMES
MIT
Cambridge, MA
moberst@mit.edu

**David Sontag**
CSAIL & IMES
MIT
Cambridge, MA
dsontag@csail.mit.edu

## Abstract

We give a method for proactively identifying small, plausible shifts in distribution which lead to large differences in model performance. These shifts are defined via parametric changes in the causal mechanisms of observed variables, where constraints on parameters yield a "robustness set" of plausible distributions and a corresponding worst-case loss over the set. While the loss under an individual parametric shift can be estimated via reweighting techniques such as importance sampling, the resulting worst-case optimization problem is non-convex, and the estimate may suffer from large variance. For small shifts, however, we can construct a local second-order approximation to the loss under shift and cast the problem of finding a worst-case shift as a particular non-convex quadratic optimization problem, for which efficient algorithms are available. We demonstrate that this second-order approximation can be estimated directly for shifts in conditional exponential family models, and we bound the approximation error. We apply our approach to a computer vision task (classifying gender from images), revealing sensitivity to shifts in non-causal attributes.

## 1 Introduction

Predictive models may perform poorly outside of the training distribution, a problem broadly known as dataset shift [Quiñonero-Candela et al., 2008]. In high-stakes applications, such as healthcare, it is important to understand the limitations of a model in advance [Finlayson et al., 2021]: given a model trained on data from one hospital, how will it perform under changes in the population of patients, in the incidence of disease, or in the treatment policy?

In this paper, our goal is to **proactively** understand the sensitivity of a predictive model to dataset shift, using only data from the training distribution. This requires domain knowledge, to specify what type of distributional changes are plausible. Formally, for a model $f(X)$ trained on data from $\mathbb{P}(X, Y)$, with loss function $\ell(f(X), Y)$, we seek to understand the loss of the model under a set of *plausible* future distributions $\mathcal{P}$. We seek to evaluate the worst-case loss over $\mathcal{P}$,

$$\sup_{P \in \mathcal{P}} \mathbb{E}_P[\ell(f(X), Y)], \tag{1}$$

and provide an interpretable description of a distribution $P$ which maximizes this objective. If the value of the worst-case loss is low, this can build confidence prior to deployment, and otherwise, examining the worst-case distribution $P$ can help identify weaknesses of the model. To illustrate, we use the following running example, inspired by Subbaswamy et al. [2021].

---

[*]Equal Contribution, order determined by coin flip. Code is available at this link.

36th Conference on Neural Information Processing Systems (NeurIPS 2022).

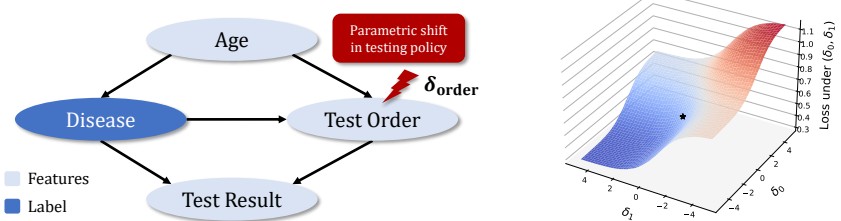

Figure 1: (Left) Causal graph for Example 1, with a shift in conditional testing rates, parameterized by $\delta_{\text{order}}$. (Right) We illustrate a shift using $s(Y; \delta_{\text{order}}) = \delta_1 \cdot Y + \delta_0(1 - Y)$, where $\delta_{\text{order}} = (\delta_0, \delta_1)$. Here we plot the (non-concave) landscape of the expected cross-entropy loss of a fixed model over distributions parameterized by $(\delta_0, \delta_1)$, with the training distribution given as the black star. Simulation details are given in Appendix A.

**Example 1** (Changes in laboratory testing). We seek to classify disease $(Y)$ based on the age $(A)$ of a patient, whether a laboratory test has been ordered $(O)$, and test results $(L)$ if a test was ordered. The performance of a predictive model may be sensitive to changes in testing policies, as the *fact that a test has been ordered* itself is predictive of disease. Figure 1 (left) gives a plausible causal relationship between variables. Let $\mathbb{P}(O = 1|A, Y) = \sigma(\eta(A, Y))$, where $\sigma$ is the sigmoid function and $\eta(A, Y)$ is the log-odds. In Figure 1 (right), we show the loss under a set of new distributions parameterized by $\delta = (\delta_0, \delta_1)$, where we modify $\mathbb{P}_\delta(O = 1|A, Y) = \sigma(\eta(A, Y) + s(Y; \delta))$ for a *shift function* $s(Y; \delta) = \delta_1 \cdot Y + \delta_0 \cdot (1 - Y)$, which modifies the log-odds of testing for both sick and healthy patients. If $\delta_0, \delta_1$ are unconstrained, the worst-case occurs when all healthy patients are tested, and no sick patients are tested.

The first challenge is to define a set of possible distributions $\mathcal{P}$ such that each distribution $P \in \mathcal{P}$ satisfies two desiderata: First, they should be *causally interpretable and simple to specify*, without placing unnecessary restrictions on the data-generating process. Second, they should be *realistic*, which often entails bounding the magnitude of the shift. We construct causally interpretable shifts by defining perturbed distributions $\mathbb{P}_\delta$ using changes in causal mechanisms, parameterized by a finite-dimensional parameter $\delta$. Our main requirement is that the shifting mechanisms follow a conditional exponential family distribution. For discrete variables, this places no restriction on $\mathbb{P}$: In Example 1, $O$ is binary and the log-odds $\eta(A, Y)$ can be any function of $A, Y$. We also demonstrate that constraining $\delta$ can ensure that shifts are realistic: The unconstrained worst-case shift in Example 1 is implausible, where all healthy patients (and no sick patients) are tested. Equation (1) becomes

$$\sup_{\delta \in \Delta} \mathbb{E}_\delta[\ell(f(X), Y)], \tag{2}$$

where $\mathbb{E}_\delta$ is the expectation in the shifted distribution $\mathbb{P}_\delta$ and $\Delta$ is a bounded set of shifts.

The second challenge is evaluation of the expected loss under shift, as well as finding the worst-case shift. Under our definition of shifts, we show that the test distribution can always be seen as a reweighting of the training distribution, allowing for reweighting approaches, such as importance sampling, to estimate the expected loss under shifts. While this is practical for some distribution shifts, for others, importance sampling can lead to extreme variance in estimation. Further, finding the worst-case shift using a reweighted objective involves maximization over a non-concave objective (see Figure 1), a problem that is generally NP-hard. We derive a second-order approximation to the expected loss under shift, and show how it can be estimated without the use of reweighting. When $\Delta$ is a single quadratic constraint, we can approximate the general non-convex optimization problem in Equation (2) with a particular non-convex, quadratically constrained quadratic program (QCQP) for which efficient solvers exist [Conn et al., 2000, Section 7]. We bound the approximation error of this surrogate objective, and show in experiments that it tends to find impactful adversarial shifts.

Our contributions are as follows:

1. We provide a novel formulation of robustness sets which are defined using parametric shifts. This formulation only require that the shifting mechanisms (i.e., conditional distributions) can be modelled as a conditional exponential family (see Section 2).

2. We derive a second-order approximation to the expected loss and provide a bound on the approximation error. We show that this translates the general non-convex problem into a particular non-convex quadratic program, for which efficient solvers exist (see Section 3).

3. In a computer vision task, we find that this approach finds more impactful shifts than a reweighting approach, while taking far less time to compute, and that the resulting estimates of accuracy are substantially more reliable (see Section 4).

## 1.1 Related Work

**Distributionally robust optimization/evaluation**: Distributionally robust optimization (DRO) seeks to learn models that minimize objectives like Equation (1) with respect to the model [Duchi and Namkoong, 2021, Duchi et al., 2020, Sagawa et al., 2020]. We focus on proactive worst-case evaluation of a fixed model, not optimization, similar to Subbaswamy et al. [2021], Li et al. [2021], but we also differ in our **definition of the set of plausible future distributions** $\mathcal{P}$, often called an "uncertainty set" in the optimization literature. Prior work often defines these sets using distributional distances (such as $f$-divergences): For instance, Joint DRO [Duchi and Namkoong, 2021] allows for shifts in the entire joint distribution (i.e., all distributions in an $f$-divergence ball around $\mathbb{P}(X, Y)$), which may be overly conservative. Marginal DRO [Duchi et al., 2020] considers shifts in a marginal distribution (e.g., $\mathbb{P}(X)$), while assuming that the remaining conditionals (e.g., $\mathbb{P}(Y \mid X)$) are fixed. However, this assumption is not applicable in all scenarios: In Example 1, for instance, this assumption does not hold for a shift in testing policy. Conditional shifts are considered in recent work that focuses on evaluation [Subbaswamy et al., 2021], using worst-case conditional subpopulations. However, choosing a plausible size of conditional subpopulation is often non-obvious. In Appendix D we give a simple lab-testing example where taking worst-case 20% conditional subpopulations corresponds to an implausible shift: Healthy patients are always tested, and sick patients never tested.

In contrast, our approach uses explicit parametric perturbations to define shifts, as opposed to distributional distances or subpopulations. In addition, our approach allows for shifts in multiple marginal or conditional distributions simultaneously: In Example 1, for instance, we could model a simultaneous change in both the marginal distribution of age $\mathbb{P}(A)$, as well as the conditional distribution of lab testing $\mathbb{P}(O \mid A, Y)$, leaving other factors unchanged.

**Causality-motivated methods for learning robust models:** Several approaches proactively specify shifting causal mechanisms/conditional distributions, and then seek to learn predictors that have good performance under arbitrarily large changes in these mechanisms [Subbaswamy et al., 2019, Veitch et al., 2021, Makar et al., 2022, Puli et al., 2022]. Other approaches use environments [Magliacane et al., 2018, Rojas-Carulla et al., 2018, Arjovsky et al., 2019] or identity indicators [Heinze-Deml and Meinshausen, 2021] to learn models that rely on invariant conditional distributions.

However, when shifts are not arbitrarily strong, causality-motivated predictors can be overly conservative. In Example 1, a model that ignores all test-related features (and only uses age as a predictor) is a particularly simple example of a causality-motivated predictor, with invariant risk over changes in testing policy. Closer to our setting is a line of work that considers bounded mechanism changes in linear causal models [Rothenhäusler et al., 2021, Oberst et al., 2021], where estimation of the worst-case loss enables learning of worst-case optimal models. Our work can be seen as extending this idea to more general non-linear causal models, where we focus on evaluation rather than optimization.

**Evaluating out-of-distribution performance with unlabelled samples**: A recent line of work has focused on predicting model performance in out-of-distribution settings, where unlabelled data is available from the target distribution [Garg et al., 2022, Jiang et al., 2022, Chen et al., 2021]. In contrast, our method operates using only samples from the original source distribution, and seeks to estimate the worst-case loss over a set of possible target distributions.

In Appendix F we give a more detailed discussion of these approaches and others.

## 2 Defining parametric robustness sets

**Notation**: Let $\mathbf{V}$ denote all observed variables, where $(X, Y) \subseteq \mathbf{V}$ for features $X$ and labels $Y$, and use $\mathbb{P}(\mathbf{V})$ to denote the probability density/mass function in the training distribution. We also refer to $\mathbb{P}$ as simply "the training distribution". $\mathbb{E}[\cdot]$ and $\text{cov}(\cdot, \cdot)$ refer to the mean and covariance in $\mathbb{P}$, and for a shifted distribution $\mathbb{P}_\delta$ (Definition 1) we use $\mathbb{E}_\delta[\cdot]$, $\text{cov}_\delta(\cdot, \cdot)$. For a random variable $Z$, we use $\mathcal{Z}$ to denote the space of realizations, and $d_Z$ for dimension e.g., $Z \in \mathcal{Z} \subseteq \mathbb{R}^{d_Z}$. For a set of random variables $\mathbf{V} = \{V_1, \ldots, V_d\}$, we use $V_i$ to denote an individual element, and use $\text{PA}_{\mathcal{G}}(V_i)$ to denote the set of parents in a directed acyclic graph (DAG) $\mathcal{G}$, omitting the subscript when otherwise clear.

We begin with a general definition of a parameterized robustness set of distributions $\mathcal{P}$.

**Definition 1.** A *parameterized robustness set around* $\mathbb{P}(\mathbf{V})$ is a family of distributions $\mathcal{P}$ with elements $\mathbb{P}_\delta(\mathbf{V})$ indexed by $\delta \in \Delta \subseteq \mathbb{R}^{d_\delta}$, with $0 \in \Delta$, where $\mathbb{P}_0(\mathbf{V}) = \mathbb{P}(\mathbf{V})$.

We give examples shortly that satisfy this general definition. To construct such a robustness set, we consider distributions $\mathbb{P}_\delta$ that differ from $\mathbb{P}$ in one or more conditional distributions (Assumption 1). We require that the relevant conditional distributions can be described by an exponential family.

**Definition 2** (Conditional exponential family (CEF) distribution). $\mathbb{P}(W|Z)$ is a conditional exponential family distribution if there exists a function $\eta(Z) : \mathbb{R}^{d_Z} \to \mathbb{R}^{d_T}$ such that the conditional probability density (for continuous $W$) or probability mass function (for discrete $W$) is given by

$$\mathbb{P}(W|Z) = g(W) \exp\left( \eta(Z)^\top T(W) - h(\eta(Z)) \right), \tag{3}$$

where $T(W)$ is a vector of sufficient statistics, $T(W) \in \mathbb{R}^{d_T}$, $g(\cdot)$ specifies the density of a base measure and $h(\eta(Z))$ is the log-partition function.

Definition 2 does not restrict $\mathbb{P}(W|Z)$ for binary/categorical $W$, and captures a wide range of distributions, including the conditional Gaussian (see Appendix B.1 for other examples). Definition 2 extends to marginal distributions where $Z = \varnothing$ and $\eta(Z)$ is a constant function.

**Example 1** (Continued). Suppose the probability of ordering a test ($O$) depends on age ($A$) and disease ($Y$), such that $\mathbb{P}(O = 1|A, Y) = \sigma(\eta(A, Y))$, where $\sigma$ is the sigmoid, and $\eta$ is an arbitrary function. Here, Definition 2 is satisfied with $W = O$, $Z = (A, Y)$, and sufficient statistic $T(O) = O$.

We now state our main assumption, where we distinguish between the terms in the joint distribution of $\mathbb{P}$ that shift, which we will need to model, and those that remain fixed, which we do not.

**Assumption 1** (Factorization into CEF distributions). Let $\mathbf{W} = \{W_1, \ldots, W_m\} \subseteq \mathbf{V}$ be a "intervention set" of variables and let

$$\mathbb{P}(\mathbf{V}) = \underbrace{\prod_{W_i \in \mathbf{W}} \mathbb{P}(W_i|Z_i)}_{\text{Conditionals that shift}} \underbrace{\prod_{V_j \in \mathbf{V} \setminus \mathbf{W}} \mathbb{P}(V_j|U_j)}_{\text{Conditionals we do not model}} \tag{4}$$

be a factorization, where $Z_i, U_j, V_j \subseteq \mathbf{V}$ are possibly overlapping (or empty) sets of variables, where $\mathbb{P}(V_j \mid \varnothing) := \mathbb{P}(V_j)$. For each $W_i$ we assume $Z_i$ is known and $\mathbb{P}(W_i|Z_i)$ satisfies Definition 2.

If $\mathbb{P}(\mathbf{V})$ factorizes according to a DAG $\mathcal{G}$, the factorization in Assumption 1 is always satisfied by $Z_i = \mathrm{PA}_{\mathcal{G}}(W_i)$. While we assume data is generated according to Equation (4), we do not require knowledge of the full distribution, but only the conditionals that shift. In Appendix B.2 we show that we can also consider shifts that extend $Z_i$ to include additional variables, subject to an acyclicity constraint. We now define parametric perturbations and give the general form of the robustness sets that we consider in this work, involving simultaneous perturbations to multiple $W_i$.

**Definition 3** (Parameterized shift functions and $\delta$-perturbations). Let $s(Z; \delta) : \mathbb{R}^{d_Z} \to \mathbb{R}^{d_T}$ be a *parameterized shift function* with parameters $\delta \in \Delta \subseteq \mathbb{R}^{d_\delta}$ which is twice-differentiable with respect to $\delta$ and which satisfies $s(Z; 0) = 0$ for all $Z$. For $\mathbb{P}(W|Z)$ satisfying Equation (3), we refer to

$$\mathbb{P}_\delta(W|Z) = g(W) \exp\left( \eta_\delta(Z)^\top T(W) - h(\eta_\delta(Z)) \right)$$

as a $\delta$-perturbation of $\mathbb{P}(W|Z)$ with shift function $s(Z; \delta)$, where $\eta_\delta(Z) := \eta(Z) + s(Z; \delta)$. Note that this differs from Equation (3) in that $\eta(Z)$ is replaced by $\eta_\delta(Z)$.

**Example 1** (Continued). A model developer may be concerned about a uniform change in testing rates across all types of patients. This can be modelled by choosing $s(Z; \delta) = \delta$, for $\delta \in \mathbb{R}$, an additive intervention on the log-odds scale. A separate change in testing rates for sick and healthy patients could instead be modeled using $s(Z; \delta) = \delta_0(1 - Y) + \delta_1 Y$, using $\delta \in \mathbb{R}^2$. This reasoning extends readily to more complex shifts (e.g., allowing for age-specific changes in testing rates, with a non-linear dependence on age), as long as $s(Z; \delta)$ remains a parametric function.

While the shift function $s(Z; \delta)$ is parametric, $\eta(Z)$ is unconstrained in Definitions 2 and 3. Note that this formulation includes multiplicative shifts $\eta_\delta(Z) = (1 + \delta)\eta(Z)$ by letting $s(Z; \delta) = \delta \cdot \eta(Z)$.

**Definition 4** (CEF parameterized robustness set). For a distribution $\mathbb{P}$ and intervention set $\mathbf{W} = \{W_1, \ldots, W_m\} \subseteq \mathbf{V}$ satisfying Assumption 1, let each $\mathbb{P}_{\delta_i}(W_i|Z_i)$ be a $\delta_i$-perturbation (Definition 3) of $\mathbb{P}(W_i|Z_i)$. Then

$$\mathbb{P}_\delta(\mathbf{V}) = \left( \prod_{W_i \in \mathbf{W}} \mathbb{P}_{\delta_i}(W_i|Z_i) \right) \left( \prod_{V_j \in \mathbf{V} \setminus \mathbf{W}} \mathbb{P}(V_j|U_j) \right)$$

is called a $\delta$-perturbation of $\mathbb{P}(\mathbf{V})$, and the robustness set $\mathcal{P}$ consists of all $\mathbb{P}_\delta$ for $\delta \in \Delta_1 \times \cdots \Delta_m$.

To estimate the expected loss under $\mathbb{P}_\delta$, we will typically[2] need to estimate $\eta(Z_i)$ for each $W_i \in \mathbf{W}$. However, we make no distributional assumptions on the remaining variables $\mathbf{V} \setminus \mathbf{W}$. This is useful in applications such as computer vision, where we do not need to restrict the generative model of images given attributes (e.g., background, camera type, etc), but can still model the expected loss under changes in the joint distribution of those attributes.

*Remark* 1 (Causal Interpretation of Shifts). If available, causal knowledge helps identify which factors in the joint distribution are subject to shifts (e.g., $\mathbb{P}(O \mid Y, A)$ in Example 1), and which remain stable. It is worth noting, however, that our methodology can be used to model any change in distribution that satisfies Assumption 1, including choices of "non-causal" factorizations and shifting factors. For example, in the context of Example 1, we could choose the factorization $\mathbb{P}(Y)\mathbb{P}(O \mid Y)\mathbb{P}(L, A \mid O, Y)$, and model a change only in the conditional $\mathbb{P}(O \mid Y)$ while keeping other factors unchanged. This shift is not interpretable as a change in causal mechanisms: The shifted distribution would imply a change in the marginal distribution of age, which should be unaffected by a real-world change in laboratory testing. Nonetheless, we can still estimate a worst-case loss over such non-causal shifts in distribution. In short, our machinery can model shifts in non-causal conditionals (for example because the causal structure is unknown), though the resulting shifted distribution is not interpretable as a plausible shift in the ground-truth data generating mechanism.

## 3 Evaluation of the worst-case loss

For a fixed predictor and loss function, we can use data from $\mathbb{P}(\mathbf{V})$ to estimate the expected loss $\mathbb{E}_\delta[\ell] := \mathbb{E}_\delta[\ell(f(X), Y)]$ for a fixed $\delta$, and estimate the worst-case loss over all $\delta$ of bounded magnitude. In Section 3.1, we show that $\mathbb{P}_\delta$ shares support with $\mathbb{P}$, suggesting the use of reweighting estimators. However, these estimators can exhibit high variance for shifts that produce large density ratios (see Appendix C.5 for an example), and maximizing a reweighted objective over $\delta$ is generally a non-convex problem. In Section 3.2 we derive an approximation to the expected loss under $\mathbb{P}_\delta$, yielding a tractable surrogate optimization problem under quadratic constraints such as $\|\delta\|_2 \leq \lambda$.

*Remark* 2. The methods here can be used with an arbitrary predictor $f$ and loss function $\ell := \ell(f(X), Y)$. We do not even require access to the original predictor $f$. Both methods here simply treat $\ell$ as a random variable in $\mathbb{P}$, for which we have samples from the training distribution.

### 3.1 Modelling shifted losses using reweighting

The shifts defined in Section 2 share common support, with the following density ratio.

**Proposition 1.** *For any $\mathbb{P}_\delta(\mathbf{V}), \mathbb{P}(\mathbf{V})$ that satisfy Definition 4, $\mathrm{supp}(\mathbb{P}) = \mathrm{supp}(\mathbb{P}_\delta)$ and the density ratio $w_\delta := \mathbb{P}_\delta / \mathbb{P}$ is given by*

$$w_\delta(\mathbf{V}) = \exp \left( \sum_{i=1}^m s_i(Z_i; \delta_i)^\top T_i(W_i) \right) \exp \left( \sum_{i=1}^m h(\eta_i(Z_i)) - h(\eta(Z_i) + s_i(Z_i; \delta_i)) \right).$$

The proof can be found in Appendix G, along with all proofs for all other claims.

**Example 1** (Continued). Suppose we perturb the probability of ordering a test $O$ given age $A$ and disease $Y$ with shift function $s(Y; \delta) = \delta_0(1 - Y) + \delta_1 Y$, independently changing the conditional probability of testing for healthy and sick patients. Here, the density ratio is given by

$$w_\delta(O, A, Y) = \exp(s(Y; \delta) \cdot O) \frac{1 + \exp(\eta(A, Y))}{1 + \exp(\eta(A, Y) + s(Y; \delta))}. \tag{5}$$

---

[2]As a special case, in Appendix C.2, we show the second-order approximation (Theorem 1) can be estimated in the case of variance-scaled mean-shifts in a conditional Gaussian without estimation of all of $\eta(Z)$.

To model the loss $\mathbb{E}_\delta[\ell]$ using data from $\mathbb{P}$, we can consider an importance sampling (IS) estimator [Horvitz and Thompson, 1952, Shimodaira, 2000], observing that $\mathbb{E}_\delta[\ell] = \mathbb{E}[w_\delta(\mathbf{V}) \cdot \ell]$. This requires estimation of the density ratio $w_\delta(\mathbf{V})$, and (given a sample $\{\mathbf{V}^j\}_{j=1}^n$ from $\mathbb{P}$) yields the estimator

$$\mathbb{E}_\delta[\ell] \approx \hat{E}_{\delta,\mathrm{IS}} := \frac{1}{n} \sum_{j=1}^n \hat{w}_\delta(\mathbf{V}^j) \ell(\mathbf{V}^j). \tag{6}$$

Equation (6) can have high variance when density ratios are large, and maximizing this equation with respect to $\delta$ is a general non-convex optimization problem, which is generally NP-hard to solve.

## 3.2 Approximating the shifted loss for exponential family models

We now propose an alternative approach for approximating the loss $\mathbb{E}_\delta[\ell]$. Recalling that $\mathbb{P}_{\delta=0} = \mathbb{P}$, we use a second-order Taylor expansion around the training distribution

$$\mathbb{E}_\delta[\ell] \approx \mathbb{E}[\ell] + \delta^\top \mathrm{SG}^1 + \tfrac{1}{2} \delta^\top \mathrm{SG}^2 \delta, \tag{7}$$

where $\mathbb{E}[\ell]$ denotes the loss in the training distribution and $\mathrm{SG}^1, \mathrm{SG}^2$ are defined as follows.

**Definition 5** (Shift gradient and Hessian). For a parametric shift satisfying Definition 1 where $\delta \mapsto \mathbb{E}_\delta[\ell]$ is twice-differentiable, we denote the *shift gradient* $\mathrm{SG}^1$ and *shift Hessian* $\mathrm{SG}^2$ as

$$\mathrm{SG}^1 := \nabla_\delta \mathbb{E}_\delta[\ell]\big|_{\delta=0} \qquad \text{and} \qquad \mathrm{SG}^2 := \nabla_\delta^2 \mathbb{E}_\delta[\ell]\big|_{\delta=0}.$$

Equation (7) is a local approximation of the loss, whose approximation error we bound in Theorem 2, with smaller approximation error for smaller shifts.[3] For $\mathbb{P}_\delta$ satisfying Definition 4, $\mathrm{SG}^1$ and $\mathrm{SG}^2$ can be computed as expectations in the training distribution, without estimation of density ratios. Recall that the conditional covariance is given by $\mathrm{cov}(A, B|C) := \mathbb{E}[(A - \mathbb{E}[A|C])(B - \mathbb{E}[B|C])|C]$.

**Theorem 1** (Shift gradients and Hessians as covariances). *Assume that $\mathbb{P}_\delta, \mathbb{P}$ satisfy Definition 4, with intervened variables $\mathbf{W} = \{W_1, \ldots, W_m\}$ and shift functions $s_i(Z_i; \delta_i)$, where $\delta = (\delta_1, \ldots, \delta_m)$. Then the shift gradient is given by $\mathrm{SG}^1 = (\mathrm{SG}_1^1, \ldots, \mathrm{SG}_m^1) \in \mathbb{R}^{d_\delta}$ where*

$$\mathrm{SG}_i^1 = \mathbb{E}\left[ D_{i,1}^\top \mathrm{cov}\left( \ell, T_i(W_i) \Big| Z_i \right) \right],$$

*and the shift Hessian is a matrix of size $(d_\delta \times d_\delta)$, where the $(i,j)$th block of size $d_{\delta_i} \times d_{\delta_j}$ equals*

$$\{\mathrm{SG}^2\}_{i,j} = \begin{cases} \mathbb{E}\left[ D_{i,1}^\top \mathrm{cov}\left( \ell, \epsilon_{T_i|Z_i} \epsilon_{T_i|Z_i}^\top | Z_i \right) D_{i,1} \right] - \mathbb{E}\left[ \ell \cdot D_{i,2}^\top \epsilon_{T|Z} \right] & i = j \\ \mathrm{cov}(\ell, \ D_{i,1}^\top \epsilon_{T_i|Z_i} \epsilon_{T_j|Z_j}^\top D_{j,1}) & i \neq j, \end{cases}$$

*where $D_{i,k} := \nabla_{\delta_i}^k s_i(Z_i; \delta_i)|_{\delta=0}$, is the gradient of the shift function for $k = 1$, and the Hessian for $k = 2$. Here, $T_i(W_i)$ is the sufficient statistic of $\mathbb{P}(W_i|Z_i)$ and $\epsilon_{T_i|Z_i} := T_i(W_i) - \mathbb{E}[T(W_i)|Z_i]$.*

Theorem 1 handles arbitrary parametric shift functions in multiple variables, but for simple shift functions in a single variable, the notation simplifies substantially, as we show in Corollary 1.

**Corollary 1** (Simple shift in a single variable). *Assume the setup of Theorem 1, restricted to a shift in a single variable $W$, and that $s(Z; \delta) = \delta$. Then $D_1 = 1$, $D_2 = 0$, and*

$$\mathrm{SG}^1 = \mathbb{E}\left[ \mathrm{cov}\left( \ell, T(W) \Big| Z \right) \right] \qquad \text{and} \qquad \mathrm{SG}^2 = \mathbb{E}\left[ \mathrm{cov}\left( \ell, \epsilon_{T|Z} \epsilon_{T|Z}^\top \Big| Z \right) \right],$$

*where $T(W)$ is the sufficient statistic of $W$ and $\epsilon_{T|Z} := T(W) - \mathbb{E}[T(W)|Z]$.*

**Example 1** (Continued). Suppose that age ($A$) follows a normal distribution with mean $\mu$ and variance $\sigma^2$, and consider a shift in the mean (without changing lab testing). We can parameterize $\mathbb{P}(A)$ as an exponential family with parameter $\eta = \mu/\sigma$ and sufficient statistic $T(A) = A/\sigma$. Here, $s(\delta) = \delta$ implies a shift in the mean of $\delta$ standard deviations $\eta_\delta = \eta + s(\delta) = (\mu + \sigma\delta)/\sigma$, and we can write that $\mathrm{SG}^1 = \mathrm{cov}(\ell, A)/\sigma$ and $\mathrm{SG}^2 = \mathrm{cov}(\ell, (A - \mathbb{E}[A])^2)/\sigma^2$.

---

[3]In Appendix C.3, we give an example of a linear-Gaussian generative model where this second-order expansion is exact, corresponding to the setting of Anchor Regression [Rothenhäusler et al., 2021].

To estimate the shift gradient and Hessian from a sample from $\mathbb{P}$, for each $i = 1, \ldots, m$ we fit models $\hat{\mu}_\ell(Z_i) \approx \mathbb{E}[\ell|Z_i]$ and $\hat{\mu}_{W_i}(Z_i) \approx \mathbb{E}[T_i(W_i)|Z_i]$ and compute residuals on these predictions, which permits estimation of the gradient/Hessian as a sample average of residuals. A detailed treatment is given in Appendix C.1. Using estimates of the gradient and Hessian, we estimate the expected loss as

$$\mathbb{E}_\delta[\ell] \approx \hat{E}_{\delta,\text{Taylor}} := \hat{\mathbb{E}}[\ell] + \delta^\top \hat{\text{SG}}^1 + \frac{1}{2}\delta^\top \hat{\text{SG}}^2 \delta. \tag{8}$$

Here, there are two sources of error: Finite-sample error, due to the estimates of $\text{SG}^1, \text{SG}^2$, as well as approximation error. The latter is bounded by the norm of $\delta$ and a term that depends on the covariance between the loss and the deviations of the sufficient statistic from its shifted mean.

**Theorem 2.** *Assume that* $\mathbb{P}_\delta, \mathbb{P}$ *satisfy the conditions of Theorem 1, with a shift in a single variable* $W$, *where* $s(Z;\delta) = \delta$. *Let* $E_{\delta,\text{Taylor}}$ *be the population Taylor estimate (Equation* (7)*) and let* $\sigma(M)$ *denote the largest absolute value of the eigenvalues of a matrix* $M$. *Then*

$$\left| \mathbb{E}_\delta[\ell] - E_{\delta,\text{Taylor}} \right| \le \tfrac{1}{2} \sup_{t \in [0,1]} \sigma\left( \text{cov}_{t\cdot\delta}(\ell, \epsilon_{t\cdot\delta,T|Z}\epsilon_{t\cdot\delta,T|Z}^\top) - \text{cov}(\ell, \epsilon_{0,T|Z}\epsilon_{0,T|Z}^\top) \right) \cdot \|\delta\|^2,$$

*where* $T(W)$ *is the sufficient statistic of* $W|Z$ *and* $\epsilon_{t\cdot\delta,T|Z} = T(W|Z) - \mathbb{E}_{t\cdot\delta}[T(W|Z)]$.

To build intuition, in Appendix C.8 we give a scenario where this bound can be simplified. In particular, we consider a "covariate shift" setting [Quiñonero-Candela et al., 2008] where $X$ is standard Gaussian, $Y = f_0(X) + \epsilon$ with a noise term independent of $X$ and we consider a shift $\delta$ in the mean of $X$. When evaluating a predictor $f(X)$ with the loss $\ell$ being the squared error, the bound in Theorem 2 depends on how the modelling error $g(X) = f_0(X) - f(X)$ behaves over the domain. In particular, the bound scales as the supremum (over $t \in [0,1]$) of $\sqrt{\text{var}(g(X + t \cdot \delta)^2 - g(X)^2)}$. As a simple corollary, if our predictor is off by an additive constant factor, $f = f_0 + C$, then the bound is zero, and the approximation is exact for any $\delta$. On the other hand, if the squared modelling error $g(X)^2$ at one point $X$ tends to be a poor predictor of the squared modelling error at another point $X + t \cdot \delta$, then this variance will be large, and the approximation will be loose.

In exchange for considering a second-order approximation of the loss, we gain two benefits: Variance reduction and tractable optimization. First, the variance of $\hat{E}_{\delta,\text{Taylor}}$ is $O(\|\delta\|^4)$ for large $\|\delta\|$, while the variance of $\hat{E}_{\delta,\text{IS}}$ can be much larger: We give a simple case in Appendix C.6 where $\text{var}(\hat{E}_{\delta,\text{Taylor}}) = O(\delta^4)$ while $\text{var}(\hat{E}_{\delta,\text{IS}}) = O(\delta^2 \exp(\delta^2))$. Second, maximizing $\hat{E}_{\delta,\text{Taylor}}$ over the set $\|\delta\| \le \lambda$ can be solved in polynomial time by exploiting the quadratic structure, while maximizing $\hat{E}_{\delta,\text{IS}}$ over the constraints is generally hard, and may be infeasible in high dimensions.

### 3.3 Identifying worst-case parametric shifts

For $\lambda > 0$, we can locally approximate the worst-case loss over all distributions $\mathbb{P}_\delta$ where $\|\delta\|_2 \le \lambda$ by finding the worst-case loss in the Taylor approximation

$$\sup_{\|\delta\|_2 \le \lambda} \mathbb{E}[\ell] + \delta^\top \text{SG}^1 + \tfrac{1}{2}\delta^\top \text{SG}^2 \delta. \tag{9}$$

Since $\text{SG}^2$ is generally not negative definite, the maximization objective is non-concave. However, this particular problem is an instance of the 'trust region problem'[4] which is well-studied in the optimization literature [Conn et al., 2000], and can be solved in polynomial time by specialized algorithms (see Pólik and Terlaky [2007, Section 8.1] for an example). This follows from the fact that strong duality holds, so that the optimal solution $\delta^*$ can be characterized in terms of the Karush-Kuhn-Tucker conditions [Boyd and Vandenberghe, 2004, Section 5.2]. For this problem, we use the `trsapp` routine from NEWUOA [Powell, 2006], as implemented in the python package `trustregion`. Depending on the application and prior knowledge, one may choose constraint sets that differ from $\|\delta\| \le \lambda$. In particular, the strong duality of Equation (9) also holds when $\|\delta\|_2 \le \lambda$ is replaced by any single quadratic constraint $\delta^\top A\delta + \delta^\top b \le \lambda$, allowing for e.g., larger shifts in some directions than in others.

---

[4]Not to be confused with the 'trust region *method*', which repeatedly solves the trust region *problem*.

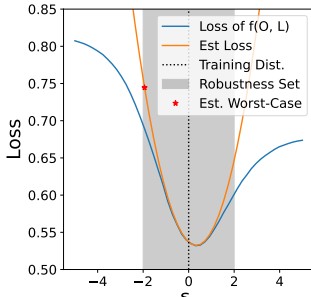 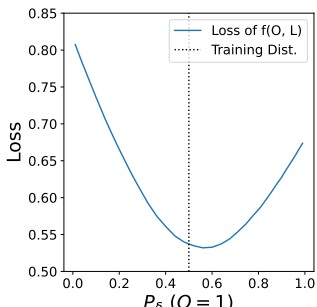

Figure 3: The blue line gives the (unobserved) cross-entropy loss under parametric shifts, plotted with respect to the parameter $\delta_0$ (left) and the resulting change in the marginal laboratory testing rate (right). We also provide the quadratic approximation (orange line), estimated using validation data, and the predicted worst-case shift (red star) for $|\delta_0| < 2$ (region in grey).

# 4 Experiments

## 4.1 Illustrative example: Laboratory testing

To build intuition, we illustrate our method in a simple generative model, similar to Example 1, where lab tests are more likely to be ordered ($O$) for sick patients ($Y$), and lab values ($L$) are predictive of $Y$.

$$Y \sim \mathsf{Ber}(0.5) \quad O|Y \sim \mathsf{Ber}(\sigma(\alpha + \beta Y)) \quad L|(Y, O = 1) \sim \mathcal{N}(\mu_y, 1)$$

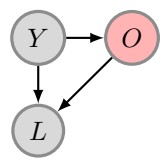

Figure 2

where $\mu_1 = 0.5, \mu_0 = -0.5$, and we initialize with $\alpha = -1$, $\beta = 2$, so that $\mathbb{P}(O = 1|Y = 0) \approx 0.27$ and $\mathbb{P}(O = 1|Y = 1) \approx 0.73$, and the marginal probability of test ordering is $\mathbb{P}(O = 1) = 0.5$. When $O = 0$, we set $L$ to a dummy value of $L = 0$. The underlying causal graph is given in Figure 2. The predictive model $f(O, L)$ is trained on data from $\mathbb{P}$ to predict $Y$ using all available features. If lab tests are not available ($O = 0$), this model predicts $Y$ based on the observed likelihood of $Y$ given $O = 0$, and otherwise uses a logistic regression model trained on cases where $O = 1$ in the training data.

**Defining a shift function:** $\mathbb{P}(O|Y)$ is a conditional exponential family with $\eta(Y) = \alpha + \beta Y$. We consider the shift function $s(Y; \delta) = \delta_0 + \delta_1 Y$, where $\delta_0$ models an overall change in testing rate, and $\delta_1$ models an additional change in the likelihood of testing sick ($Y = 1$) patients.

**Estimating the impact of shift using quadratic approximation:** To start, we keep $\delta_1 = 0$ fixed and vary only $\delta_0$, which uniformly increases or decreases testing. In Figure 3, we show the ground-truth cross-entropy loss of $f(O, L)$ under perturbed distributions $\mathbb{P}_{\delta_0}$. We observe that the **direction** of the shift matters: In Figure 3, the model performance slightly increases under a small increase in testing rates, but degrades if testing increases too much; moreover, the loss under shift is generally asymmetric, as a decrease hurts more than an increase in testing. In Figure 3 (left), we demonstrate the use of the quadratic approximation described in Section 3.2. For illustration, we consider a robustness set of $\delta_0 \in [-2, 2]$, and see that the predicted worst-case shift coincides with the actual worst-case shift, and that the quadratic approximation is accurate for smaller values of $\delta$.

In Appendix D, we allow both $\delta_0$ and $\delta_1$ to vary, and compare our approach to that of worst-case $(1 - \alpha)$ conditional subpopulation shifts [Subbaswamy et al., 2021]. In the context of this example, we demonstrate that for any $1 - \alpha < 0.27$, the worst-case conditional subpopulation loss is achieved by having all healthy patients get tested, and no sick patients get tested. We contrast this with an iterative approach to designing constraints that is made possible by considering parametric shifts, where end-users can restrict the degree to which the shift differs across sick and healthy populations.

## 4.2 Detecting sensitivity to non-causal correlations

A predictive model may pick up on various problematic dependencies in the data that may not remain stable under dataset shift. To understand the impact of these dependencies, a model user may wish to understand which changes in distribution pose the greatest threats to model performance, and to measure the impact of these changes. To illustrate this use-case, we make use of the CelebA dataset

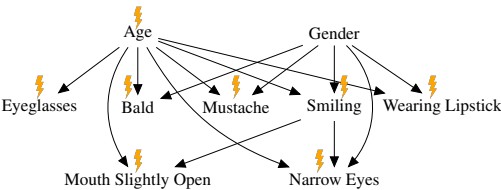

Figure 4: Causal graph over attributes in the synthetic CelebA dataset, where lightning bolts indicate changes in mechanisms. All of these attributes are causal parents of the image $X$ (not shown here), which is generated by a GAN conditioned on these attributes.

[Liu et al., 2015], which contains images of faces and binary attributes (e.g., glasses, beard, etc.) encoding several features whose correlations may be unstable (e.g., the relation between gender and being bald). We consider the task of predicting gender ($Y$) from images of faces ($X$), and assess sensitivity to a shift in the distributions of attributes ($\mathbf{W}$).[5]

**Setup**: To obtain ground-truth shifts in distribution, we generate synthetic datasets of faces using CausalGAN [Kocaoglu et al., 2018], trained on the CelebA data. We simulate attributes following the causal graph in Figure 4, and then simulate images from the GAN conditioned on those attributes. We draw a training sample from this distribution $\mathbb{P}$, and fit a gender classifier $f(X)$ using the image data alone, by finetuning a pretrained ResNet50 classifier [Hu et al., 2018]. Each attribute $W_i$ is binary, so we consider shifts in the log-odds $\eta_i(Z_i)$ of each attribute $W_i$ given parents $Z_i$. Here, we use a maximally flexible shift function $s_i(Z_i; \delta_i) = \sum_{z \in \mathcal{Z}_i} \delta_{i,z} \mathbf{1}\{Z_i = z\}$, such that for $Z_i \in \{0,1\}^k$ there are $2^k$ parameters. Across all intervened variables, $\delta \in \mathbb{R}^{31}$. Due to the synthetic nature of our setup, we can simulate from $\mathbb{P}_\delta(X, \mathbf{W}, Y)$ to evaluate the ground-truth impact of this shift, simulating first from the shifted attribute distribution, and then simulating images from the GAN conditional on those attributes. We use the 0/1 loss $\ell = \mathbf{1}\{f(X) \neq Y\}$, and constrain $\delta$ by $\|\delta\|_2 \leq \lambda = 2$.

**Comparing importance sampling and Taylor across multiple simulations**: We simulate $K = 100$ validation sets from $\mathbb{P}$, in each estimating the worst-case shifts $\delta_{\text{Taylor}}$ (via the approach in Section 3.3) and $\delta_{\text{IS}}$, where the latter corresponds to minimizing $\hat{E}_{\delta,\text{IS}}$ using a standard non-convex solver from the scipy library [Virtanen et al., 2020]. We simulate ground truth data from $\mathbb{P}_{\delta_{\text{IS}}}$ and $\mathbb{P}_{\delta_{\text{Taylor}}}$, to compare the two shifts. First, we demonstrate that the Taylor approach finds more impactful shifts, when searching over the space of small, bounded shifts considered here. In Table 1 (right), we compare the average drop in accuracy using the Taylor shifts (3.8%) and the IS shifts (2.2%). In Figure 5 (right) we plot the differences in test accuracy $\mathbb{E}_{\delta_{\text{Taylor}}}[\mathbf{1}\{f(X) = Y\}] - \mathbb{E}_{\delta_{\text{IS}}}[\mathbf{1}\{f(X) = Y\}]$, where the Taylor approach finds a more impactful shift in 96% of cases. Second, the Taylor approach has an average run-time of $0.01s$, versus $2.14s$ for the IS approach. Third, when **only** used to evaluate the shift $\delta_{\text{Taylor}}$, the IS estimator is comparable to the Taylor estimator, with a near-identical average bias (shown in Table 1 (right)) and RMSE (0.0191 and 0.0192 respectively). Finally, however, in Table 1 (right) we observe that $\hat{E}_{\delta_{\text{IS}},\text{IS}}$ is strongly biased in predicting $\mathbb{E}_{\delta_{\text{IS}}}$, yielding a mean absolute prediction error (MAPE) of 0.069 (not shown in the table). This can be contrasted with a MAPE of 0.015 when using $\hat{E}_{\delta_{\text{Taylor}},\text{Taylor}}$ to predict $\mathbb{E}_{\delta_{\text{Taylor}}}$. This may suggest that optimizing the IS objective is prone to "overfitting", choosing a sub-optimal $\delta$ from a region of the search space that has high variance. Here, where $\lambda = 2$, the drop in accuracy is relatively mild for the shifts found by both approaches. In Appendix E.4 we show that larger values of $\lambda$ correspond to more substantial drops in accuracy (e.g., an average drop of 23% for $\lambda = 8$ using the Taylor approach).

**Examining a single shift**: To illustrate the type of shift found by our approach, we consider the $\delta_{\text{Taylor}}$ (over the $K$ runs) which yields the $\mathbb{P}_\delta$ with median test accuracy. We display the largest components of that $\delta$ in Table 1 (left). Among others, this shift entails a 5% increase in the probability of an older woman being bald, and a 5% decrease in the probability of a young woman wearing lipstick. This suggests that the learned classifier $f$ relies on these associations in the images for prediction. We validate that this shift leads to a decrease in accuracy of around 3.8%, using simulated data from $\mathbb{P}_\delta$. To validate that this drop in accuracy is a non-trivial occurrence, we simulate $K = 400$ random shifts

---

[5]We do not endorse gender classification as an inherently worthwhile task. Nonetheless, gender classification is commonly studied in the context of understanding the implicit biases of machine learning models [Buolamwini and Gebru, 2018, Schwemmer et al., 2020], and we consider the task with that context in mind.

Table 1: (Left) Top 5 components (by magnitude) of the example shift vector $\delta \in \mathbb{R}^{31}$ where $\mathbb{P}$ and $\mathbb{P}_\delta$ denote conditional probabilities. The full example shift vector can be found in Appendix E.2. (Right) Taylor and IS estimates vs. true accuracy for the $\delta_{\text{Taylor}}$ found by the Taylor approach, and IS estimate vs. true accuracy for the $\delta_{\text{IS}}$ found by the IS approach. Averages are taken over 100 simulations.

| Conditional | | $\delta_i$ | $\mathbb{P}$ | $\mathbb{P}_\delta$ |
|---|---|---|---|---|
| Bald | \| Female, Old | 0.899 | 0.047 | 0.109 |
| Bald | \| Male, Young | -0.800 | 0.378 | 0.214 |
| Bald | \| Male, Old | -0.680 | 0.622 | 0.455 |
| Wearing Lipstick | \| Female, Young | -0.618 | 0.924 | 0.868 |
| Wearing Lipstick | \| Female, Old | -0.543 | 0.953 | 0.921 |

| Metric | Example $\delta$ | Avg. |
|---|---|---|
| Original acc. ($\mathbb{E}[\mathbf{1}\{f(X) = Y\}]$) | | 0.912 |
| Acc. under Taylor shift ($\mathbb{E}_{\delta_{\text{Taylor}}}[\mathbf{1}\{f(X) = Y\}]$) | 0.874 | 0.874 |
| IS est. of acc. under Taylor shift ($\hat{E}_{\delta_{\text{Taylor}},\text{IS}}$) | 0.829 | 0.863 |
| Taylor est. of acc. under Taylor shift ($\hat{E}_{\delta_{\text{Taylor}},\text{Taylor}}$) | 0.844 | 0.863 |
| Acc. under IS shift ($\mathbb{E}_{\delta_{\text{IS}}}[\mathbf{1}\{f(X) = Y\}]$) | | 0.889 |
| IS est. of acc. under IS shift ($\hat{E}_{\delta_{\text{IS}},\text{IS}}$) | | 0.821 |

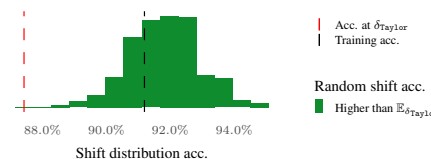

Figure 5: (Left) Model accuracy at randomly drawn shifts. (Right) Difference in accuracy in the worst-case shifts identified by Taylor and importance sampling approaches. The Taylor method identifies a more adversarial shift than importance sampling in 96% of simulations (green).

$\delta_k$ where $\|\delta_k\| = \lambda$ and evaluate the model accuracy in $\mathbb{P}_{\delta_k}$ (Figure 5, left). As expected, the chosen $\delta$ yields a lower accuracy (red line) than all of the random shifts.

## 5 Conclusion

We argue for considering parametric shifts in distribution, to evaluate model performance under a set of changes that are interpretable and controllable. For parametric shifts in conditional exponential family distributions, we derive a local second-order approximation to the loss under shift. This approximation enables the use of efficient optimization algorithms (to find the worst-case shift), and empirically provides realistic estimates of the resulting loss. In a computer vision task, this approach finds more impactful shifts (in far less time) than optimizing a reweighted objective, and the estimates of shifted accuracy under the chosen shift are substantially more reliable.

Of course, our method is not without limitations. Our definition of parametric shifts and resulting approximation relies on the relevant mechanisms $\mathbb{P}(W|Z)$ being a conditional exponential family, and that the relevant variables are observed. As illustrated in our experiments, this can be used to model changes in the causal relationships **between** attributes of an image, but does not immediately extend to modelling changes in the distribution of images given a fixed set of attributes. As with any method that provides worst-case evaluation, there is potential for misuse and false confidence: If the specified shifts fail to capture important real-world changes, the resulting worst-case loss may be overly optimistic and misleading. Even if used correctly, our approach examines a narrow measure of model performance, and a small worst-case error should not be used to claim that a model is free of problematic behavior. For example, implicit dependence on certain attributes (e.g., race in medical imaging [Banerjee et al., 2021]) may be problematic based on ethical grounds, even if it does not lead to major issues with predictive performance under small shifts in distribution.

### Acknowledgements

We thank Jonas Peters, Tommi Jaakkola, Chandler Squires, and Stefan Hegselmann for helpful feedback and discussion, and Irene Chen and Christina X Ji for providing comments on an earlier draft. MO and DS were supported in part by Office of Naval Research Award No. N00014-21-1-2807. NT was supported by a research grant (18968) from VILLUM FONDEN.

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
