# OpenReview forum: "Evaluating Robustness to Dataset Shift via Parametric Robustness Sets"
_NeurIPS.cc/2022/Conference — NeurIPS 2022 Accept_

### Official Review · Reviewer_mdu8 · 2022-06-25

**Rating:** 5
**Confidence:** 2
**Soundness:** 2 fair
**Presentation:** 1 poor
**Contribution:** 2 fair

**Summary:**

The paper considers the problem of identification and evaluation of potential harmless data shifts. The paper proposes a parametric robustness set and proposes a local second-order approximation to efficiently solve the worst-case problem

**Questions:**


Under which scenarios, the local second-order approximation performs well?

**Limitations:**

The definition of CEF parametrized robustness set seems restrictive.



**Strengths And Weaknesses:**

The problem of proactively understanding the policy's performance under distributional shifts is interesting to me.

However, honestly, I feel this paper is hard to understand, perhaps I am not very familiar with the language used in this community.
1. For Example 1, are A, and Y are allowed to change? There is a $\delta_{age}$ in the caption of Figure 1, but it seems that the authors never use them.
2. The author $\mathbb{P}(V)$ to denote a distribution, which is not commonly used. I suggest the authors rigorous define everything using probability language, e.g. define the probability space {$ \Omega,\mathcal{F},P $}.
3. I am confused with Equation (4) in Assumption 1. Why there is no $\mathbb{P}(Z_i)$ and $\mathbb{P}(U_j)$.
4. I think the authors would better explain what shifts, intervention, and robust mean in this context.
5. I did not quite get the intuition behind the CEF parametrized robustness set? Why is it a natural and good choice? The authors should better motivate.
6. Are the variables $W$ the only variables that only to be intervened (shifted)? How about the variables Z,V,U?
7. I think Proposition 1 implicitly assumes the distributions of Z do not change.


Overall I find this paper needs to be further polished. And this paper at least in its current version will create difficulties for the general audience of NeurIPS to understand.

---

> ### Author Response · Authors · 2022-08-02
> **Response (Part 2)**
>
> **I did not quite get the intuition behind the CEF parametrized robustness set? Why is it a natural and good choice? The authors should better motivate.**
>
> It is a natural choice for restricting ourselves to plausible & interpretable shifts in distribution, as we discuss throughout in the context of Example 1.  Please see our response to reviewer 93gd in particular, regarding the generality of this notion of shift.
>
> In Appendix D we give an illustration for why this is a more natural choice than some alternatives, by contrasting against the worst-case $(1 - \alpha)$ conditional subpopulation shift.  In particular, we demonstrate that it allows us to avoid taking the worst-case over implausible distributions, like only testing healthy patients in the context of Example 1.
>
> **Are the variables W the only variables that only to be intervened (shifted)? How about the variables Z,V,U?**
>
> Only the variables $W$ are shifted, but the set $W$ can overlap with the variables in $Z, V, U$.  Note that $Z$ and $U$ are sets of sets of variables.
>
> For instance, in the factorization $P(A) P(Y | A) P(O | A, Y) P(L | Y, O)$, for a shift in age + lab orders, we would have $W_1 = A, Z_1 = \varnothing, V_1 = Y, U_1 = (A), W_2 = O, Z_2 = (A, Y), V\_2 = L$ and $U_2 = (Y, O)$, so that $W = (A, O), Z = (\varnothing, (A, Y)), V = (Y, L)$, and $U = ((A), (Y, O))$.
>
> Note that the variable $A$ appears in the set $W$, but also in one of the sets in $Z$, and one of the sets in $U$.
>
> **I think Proposition 1 implicitly assumes the distributions of Z do not change.**
>
> See above.
>
> **Under which scenarios, the local second-order approximation performs well?**
>
> We provide some intuition via an example (in Appendix C.3) where the approximation is exact, which occurs in linear-Gaussian models when using a squared error.
>
> We have also added an additional example as Appendix C.8 in the revision, that illustrates how the bound can be simplified in a particular “covariate shift” setting, where $Y = f_0(X) + \epsilon$ and the distribution of $X$ is Gaussian, with a shift in the mean.  In this case, as one would expect, the approximation error depends on the error $f_0(x) - f(x)$, where $f(x)$ is our predictor, and different assumptions on how the modeling error $g(x) = f_0(x) - f(x)$ behaves over the domain.  In particular, the error in the approximation scales as the supremum (over $t \in [0, 1]$) of $\sqrt{var(g(X + t * \delta)^2 - g(X)^2)}$.  As a simple example, if $g(x) = f_0(x) - f(x)$ is a constant, i.e. if our predictor differs from the true function by an additive constant factor, the bound is $0$.

---

> > ### Comment · Reviewer_mdu8 · 2022-08-08
> > **Thanks for the response**
> >
> > I increased my score as the authors revise the manuscript and clarify some of my questions. However, I still think the paper in its current draft is hard for the general audience of Neurips to understand.

---

> ### Author Response · Authors · 2022-08-02
> **Response (Part 1)**
>
> Thank you for the questions, and we apologize for the confusion. Please see our clarifications below, but let us know if there are any remaining points of confusion during discussion.
>
> **For Example 1, are A, and Y are allowed to change? There is a d_age in the caption of Figure 1, but it seems that the authors never use them.**
>
> Apologies for the confusion.  Our goal was to convey that both $P(A)$ and $P(O | A, Y)$ can change simultaneously in our framework.  However, for most of the examples, we only discuss a shift in the latter for simplicity, and do not discuss shifts in $P(A)$ until the example block on line 200 (in the revision) later in the main text.  In the revision, we have removed $d_{age}$ from Figure 1 for clarity.
>
> **The author P(V) to denote a distribution, which is not commonly used. I suggest the authors rigorous define everything using probability language, e.g. define the probability space {Omega, F, P}**
>
> In the notation section, we have changed “we let $\mathbb{P}(V)$ denote the training distribution” to “we use the notation $\mathbb{P}(V)$ to refer to the probability density/mass function in the training distribution, and will also refer to $\mathbb{P}(V)$ as ‘the training distribution’”, which we agree could be otherwise misunderstood. To clarify, we consider $d$ random variables $V_1, \ldots, V_d$ and use $V$ to denote the collection of them.
>
> The triplet $\{\Omega, \mathbb{F}, P\}$ will depend on the concrete distributions in question, e.g., whether or not they are discrete or continuous. Our main motivation is considering distributions on $\mathbb{R}$ and $\mathbb{N}$, where the sigma-algebra will typically just be the Borel sigma-algebra (which coincides with the power set on $\mathbb{N}$), and we therefore prefer to keep this implicit to avoid confusion for readers unfamiliar with measure theory.
>
> **I am confused with Equation (4) in Assumption 1. Why there is no P(Z_i) and P(U_j)**
>
> As clarified below the equation, each set $Z_i$, $U_j$, $V_j$ are possibly overlapping.  Consider a factorization of some joint distribution $P(A, B, C) = P(A | B, C) P(B | C) P(C)$, where $W = \{A\}$.  This factorization can be written in this form where $W_1 = A, Z_1 = \{B, C\}, V_1 = B, U_1 = \{C\}, V_2 = C, U_2 = \varnothing$.  In the revised version, we have clarified that when the conditioning set is empty, this notation corresponds to the marginal distribution (e.g., in the example above where $V_2 = C$ and $U_2 = \varnothing$, our convention is that $P(V_2 \mid U_2) = P(C)$).
>
> **I think the authors would better explain what shifts, intervention, and robust mean in this context.**
>
> Here, a “shift” is a change in one or more conditional distributions, and the term “intervention” refers to the fact that, when these conditional distributions can be written as $P(W_i | PA(W_i))$ (i.e., the conditioning set contains the causal parents of $W_i$), they can be interpreted as interventions on “causal mechanisms” (see Def. 1 in Section 3.2 of [1]).
>
> Our notion of being “robust” is equivalent to having low worst-case loss over a set of distributions, as in Equation (1), which we refer to here as a “robustness set” but which is often referred to as an “uncertainty set” of distributions (see Section 2 of [2]) in the distributionally robust optimization literature.
>
> [1] Subbaswamy, A., Chen, B., & Saria, S. (2022). A unifying causal framework for analyzing dataset shift-stable learning algorithms. Journal of Causal Inference, 10(1), 64–89.
>
> [2] Duchi, J. C., & Namkoong, H. (2021). Learning models with uniform performance via distributionally robust optimization. The Annals of Statistics, 49(3), 1378–1406.

---

### Official Review · Reviewer_93gd · 2022-07-09

**Rating:** 6
**Confidence:** 4
**Soundness:** 3 good
**Presentation:** 3 good
**Contribution:** 2 fair

**Summary:**

The paper aims to tackle the problem of model evaluation by understanding the sensitivity of a predictive model to dataset shift using only data from the training distribution. The authors consider parametric shifts with Conditional Exponential Family (CEF) under a set of changes that are interpretable and controllable. For CEF distributions, authors derive a local second-order approximation to the loss (Theorem 1). This approximation enables the use of efficient optimization algorithms to find the worst-case shift. Authors also provide error bounds to their approximation (Theorem 2). On CelebA classification task, the authors show the efficacy of the proposed framework.

**Questions:**

While the motivation of the paper is very interesting and the proposed framework is novel, I have several suggestions/questions:

- How to decide the bounds on $\delta$, i.e., robustness set $\mathcal{P}$ is unclear. Could the authors elaborate a bit more here?
- To tackle the variance issue in the importance weights, in Eq. 6, did the authors try doubly robust estimators [1]? Maybe this can already improve the numbers in Table 2 (right) for IM.
- As mentioned in the weaknesses above, could the authors explain how general their considered CEF framework is?
- How does the framework considered here compare with Mandoline's framework [2]? It would be great to see the `pros and cons' of the CEF framework with the Mandoline framework, in terms of coverage of model evaluation settings, meta-information needed from humans, and underlying assumptions?
- How does the accuracy estimation in CelebA experiment stand with other heuristic methods, e.g., GDE [3], ATC[4].


[1]  Reddi et al. 2014. Doubly Robust Covariate Shift Correction. https://www.cs.cmu.edu/~sjakkamr/papers/doublyrobust.pdf.

[2] Chen et al. 2021. Mandoline: Model Evaluation under Distribution Shift https://arxiv.org/abs/2107.00643

[3] Jiang et al. 2022. Assessing Generalization of SGD via Disagreement. https://arxiv.org/abs/2106.13799.

[4] Garg et al. 2022. Leveraging Unlabeled Data to Predict Out-of-Distribution Performance. https://arxiv.org/abs/2201.04234

**Limitations:**

The authors provide a summary of the limitations of their work.

**Strengths And Weaknesses:**

**Strength**:

- The problem considered in the paper is very interesting and timely. Understanding model performance beyond iid validation test is important prior to deployment. This can also enable examining the worst-case distribution P which can help identify weaknesses of the model.
- Approximation of the loss with the CEF with second-order Taylor expansion is interesting. Theorem 1 derives an expression for SG$_i^1$ and Theorem 2 provides an error bound for the approximation error in population.
- The running Example 1 is illustrative. Connecting back to it at many places helps in understanding the proposed framework and formulation


**Weakness**:
- The paper could do a better job in explaining what all kinds of distribution shifts are captured with Definition 3. Again the example in Section 4.1 is illustrative and interesting but overall, it is not clear how general the considered CEF framework is?
- Experimental evaluation of the proposed framework is very slim and a major weakness of the paper. While I acknowledge the theoretical contribution and novelty of the paper, as mentioned before, the generality of the CEF framework is unclear and experimental validation is lacking. This is the main reason for my score. I am open to changing my score if the authors could comment more on this point or if I share any misunderstanding.
  - (Minor) In the CelebA experiment, how can a user identify which variables to intervene upon and how to decide their bounds, i.e., the problematic dependencies (for which model performance changes considerably under dataset shift) in the data?
  - (Minor) Authors use Causal GAN to simulate distribution shifts, how reliable is even the ground truth evaluation in this framework? While I appreciate the details in Appendix D with some pictures of the generated example, a commentary on the generated dataset would be great.
  - As mentioned in the limitations section, specified shifts can fail to capture important real-world changes and hence, the resulting worst-case loss may be overly optimistic and misleading. Can authors comment on this more in the context of CelebA experiment where the worst-case shift identified only has a drop of ~3 percentages in accuracy?
- (Minor) While authors acknowledge the sources of error due to finite samples, complimenting Theorem 2 with errors due to finite-sample approximation would be insightful.
- (Minor) Paper is a bit hard to understand at places. For example,
  - Definition 3 gives the parametric family of shifts considered in the paper. The changes w.r.t. Definition 2 could be reminded to the reader which in particular is in the $\eta_\delta$ function.
- (Minor) Writing of the paper could also be improved. For example,
  -  A lot of references to "these" and "them" are used where the contextualization is a bit ambiguous and left for interpretation. E.g., abstract line 2-5.

---

> ### Author Response · Authors · 2022-08-02
> **Response (Part 3)**
>
> **How to decide the bounds on delta, i.e., robustness set $\mathcal{P}$ is unclear. Could the authors elaborate a bit more here?... In the CelebA experiment, how can a user identify which variables to intervene upon and how to decide their bounds, i.e., the problematic dependencies (for which model performance changes considerably under dataset shift) in the data?**
>
> In the CelebA experiment, we demonstrate one approach for choosing which variables to intervene on: We allow $\delta$ to influence all the attribute variables (except for the label), and then let the algorithm discover which shifts yield the largest drops in performance.
>
> For choosing bounds on $\delta$, in many cases users will have some prior knowledge.  For instance, in the lab test ordering example, one may decide to consider shifts where the marginal probability of ordering a test does not change by more than 20%.  In Appendix D.3, we give an example where this type of constraint can be translated directly into bounds on $\delta$ (lines 1047-1052 of the revision).  More broadly, however, choosing the “right” constraint is a problem faced by many other methods that deal with worst-case distribution shifts, which require a user-specified hyperparameter to control the size of a worst-case subpopulations [1, 2] or the size of an $f$-divergence ball of possible distributions [3, 4].
>
> We see our approach as being most useful when used in an iterative / exploratory fashion, to understand the sensitivity of a model: Starting with a broad class of possible shifts, and then (if the resulting change is not plausible from the perspective of a domain expert) iteratively restricting the set of shifts (either by setting tighter bounds on $\delta$ or removing / restricting some shift functions).  We give an illustrative example of such a process in Appendix D.3.
>
> [1] Duchi, J., Hashimoto, T., & Namkoong, H. (2020). Distributionally robust losses for latent covariate mixtures. arXiv Preprint arXiv:2007.13982.
>
> [2] Subbaswamy, A., Adams, R., & Saria, S. (2020). Evaluating Model Robustness and Stability to Dataset Shift. Proceedings of the 24th International Conference on Artificial Intelligence and Statistics.
>
> [3] Duchi, J. C., & Namkoong, H. (2021). Learning models with uniform performance via distributionally robust optimization. The Annals of Statistics, 49(3), 1378–1406.
>
> [4] Ahmadi-Javid, A. (2012). Entropic Value-at-Risk: A New Coherent Risk Measure. Journal of Optimization Theory and Applications, 155(3), 1105–1123.
>
> **Experimental evaluation of the proposed framework is very slim and a major weakness of the paper. While I acknowledge the theoretical contribution and novelty of the paper, as mentioned before, the generality of the CEF framework is unclear and experimental validation is lacking.**
>
> We have updated our experiment in Section 4.2 with more details on the performance of our method and importance sampling, as discussed in our general comment.  We also point the reviewer to the experiment in Appendix C.5. and Figure 8, where we employ our method in a simulated setting with both binary and continuous variables. In particular, Figure 8 shows a scenario where the shift in a binary variable is so sufficiently simple that importance sampling is able to correctly predict the impact, but completely fails to capture the mean shift in a Gaussian variable. While our method exhibits more bias than IS in the binary variable, it predicts the shifted loss much more accurately in the Gaussian distribution.
>
> **(Minor) While authors acknowledge the sources of error due to finite samples, complimenting Theorem 2 with errors due to finite-sample approximation would be insightful.**
>
> We agree that this type of result could be insightful, but see this as outside the scope of the author's response.  Even understanding the precise finite-sample behavior of importance sampling (rather than using bounds or asymptotics) is non-trivial.
>
> We do provide some intuition on the relative variance of each approach in the paragraph following Theorem 2, for a simple example that we detail in Appendix C.6.
>
> **(Minor)... Definition 3 gives the parametric family of shifts considered in the paper. The changes w.r.t. Definition 2 could be reminded to the reader which in particular is in the eta_delta function…. A lot of references to "these" and "them" are used where the contextualization is a bit ambiguous and left for interpretation. E.g., abstract line 2-5.**
>
> We appreciate the feedback, and have incorporated these points into the revision.

---

> > ### Comment · Reviewer_93gd · 2022-08-06
> > **Thank you for your responses**
> >
> > I thank the authors for their detailed comments and clarification. I have increased my score to 6 based on the clarifications provided by the authors.
> >
> > A few minor questions: I am not fully sure if I understand the results in App E.4. Mainly the numbers on what is the predicted accuracy drop. To the best of my understanding authors only provide the actual accuracy drop numbers.
> >
> > While I am trying to parse the added results more, is it possible for the authors to highlight the key changes in the paper with a different color?

---

> > > ### Author Response · Authors · 2022-08-07
> > > **Thank you**
> > >
> > > Thank you for engaging in the discussion!  Some clarifications:
> > >
> > > **Estimates under different values of $\lambda$**: Apologies for the confusion here. Yes, the numbers in Appendix E.4 only refer to the ground-truth impact of the shifts found by the Taylor approach (under larger values of $\lambda$).  This was tailored to your original question, which seemed to be "is it possible to find a more impactful shift under this framework". Given your follow-up about the predicted accuracy drop, we ran those numbers as well, while also including results for $\lambda = 6$.
> > > * For the $\delta_{\text{Taylor}}$ found by the Taylor approach for $\lambda = 4$, the ground truth accuracy across the simulations is 81.2% on average (a 10% drop), and the IS and Taylor estimators predict an accuracy of 79.5% and 79.8% on average, respectively.
> > > * For $\lambda = 6$, the ground truth is 73.6% (a 17% drop), and IS/Taylor predict 71.5% and 71.1%.
> > > * For $\lambda = 8$, the ground truth is 68.1% (a 23% drop), and IS/Taylor predict 65.8% and 60.1% respectively.
> > >
> > > This supports the utility of the approach "use Taylor to find a shift, and then use IS to evaluate", as mentioned in our response to reviewer Qr2b. For $\lambda = 2$, the bias of this approach is ~1% (predicting 86% vs ground truth of 87%), and for $\lambda = 8$, the bias of this approach is still only ~2% (predicting 66% versus 68%).  We have added this additional detail to Appendix E.4 in the revision.
> > >
> > > **Making changes clear**: We have uploaded a new version that highlights the key changes / additions in Section 4.2 in blue, as well as highlighting the new appendix sections.  Beyond that, our general comment lists out all the key changes, where the changes outside of Section 4.2 are small clarifications, not new results.

---

> ### Author Response · Authors · 2022-08-02
> **Response (Part 2)**
>
> **As mentioned in the limitations section, specified shifts can fail to capture important real-world changes and hence, the resulting worst-case loss may be overly optimistic…can authors comment on this more in the context of CelebA experiment where the worst-case shift identified only has a drop of ~3 percentages in accuracy?**
>
> The drop in accuracy is a function of the constraints $||\delta|| \leq \lambda$, and the choice of $\lambda$.  In light of this comment, we conducted the following experiment, and have added this detail to Appendix E.4 in the revision (and will add to the main text in the camera ready if accepted, given the additional page):  Over 100 synthetic training datasets, we repeated our experimental setup (finding a shift $\delta$ using the Taylor approach, and evaluating the loss under $P_{\delta}$ via simulation).  For $\lambda = 4$, the average drop in accuracy was 10%, and for $\lambda = 8$, the average drop in accuracy was 23%.
>
> **How does the framework considered here compare with Mandoline's framework [2]? It would be great to see the `pros and cons' of the CEF framework with the Mandoline framework, in terms of coverage of model evaluation settings, meta-information needed from humans, and underlying assumptions?... How does the accuracy estimation in CelebA experiment stand with other heuristic methods, e.g., GDE [3], ATC[4]...did the authors try doubly robust estimators [1]?**
>
> The first three approaches (Mandoline, GDE, and ATC) tackle a very different problem from ours.  They seek to estimate the loss under a *particular* target distribution, given access to *unlabelled samples* from that target distribution.  In contrast, our method operates using only samples from the original source distribution, and seeks to estimate the worst-case loss over a set of possible future distributions.  We have added this clarification to the “comparisons to other methods” discussion in Appendix B.4 in the revision (which will be included in the main text of the camera ready, if the paper is accepted).
>
> The fourth approach (citation [1]) is also not directly applicable in our setting, for two reasons:  First, similar to the above, it assumes access to unlabelled samples from a specific target distribution, and seeks to estimate loss under that distribution.  Second, it assumes that $P(Y \mid X)$ is constant between the two distributions (Assumption 1 of the cited work):  In our setting, this does not hold, as the distributions we consider directly modify the distribution of attributes given the label.

---

> ### Author Response · Authors · 2022-08-02
> **Response (Part 1)**
>
> Thank you for the thorough review, and for raising these questions.  Hopefully our response can clarify some points:
>
> **The paper could do a better job in explaining what all kinds of distribution shifts are captured with Definition 3. Again the example in Section 4.1 is illustrative and interesting but overall, it is not clear how general the considered CEF framework is... could the authors explain how general their considered CEF framework is?**
>
> Our framework captures shifts in any finitely supported distribution, including binary and categorical distributions, since these are exponential families.  If all variables are discrete, then the entire joint distribution can be seen as a single categorical distribution, and our framework can capture any change in the distribution which does not alter the support.
>
> Our framework also handles shifts in many continuous distributions, but it does not cover all possible shifts.  For instance, for a variable $W$ where $P(W \mid Z)$ is conditionally Gaussian with mean $\mu(Z)$ and variance $\Sigma(Z)$, we only allow for shifts in distribution that change this mean and variance, but we do not allow for shifts that yield a non-Gaussian distribution. Likewise we do not cover shifts in non-exponential-family distributions. In Section 4.2, for instance, we do not consider shifts in the distribution of images given attributes, but only shifts in the joint distribution of the attributes themselves. These restrictions are consequences of the goal of finding interpretable shifts – if a distribution could shift arbitrarily, we may not be able to quantify in which way the distribution shifted.
>
> In addition, it is worth noting that our framework is not restricted to common notions of shift like covariate shift ($P(X)$ changes but $P(Y \mid X)$ is fixed) or label shift ($P(Y)$ changes while $P(X \mid Y)$ is fixed).  Even in the simple example of Section 4.1, both $P(X)$ and $P(Y \mid X)$ are changing, but the problem remains tractable because they are changing in a restricted fashion, not in an arbitrary way.
>
> **Authors use Causal GAN to simulate distribution shifts, how reliable is even the ground truth evaluation in this framework?**
>
> Our experiments evaluate how well our methods perform *at the task they are designed to do*:  Finding a worst-case distribution within a certain set, and predicting the loss under that new distribution.  In order to evaluate both of these points, we needed a reliable way to generate data from a particular set of novel distributions, which is why we used a simulation-based approach.
>
> With this in mind, our experiments give reliable estimates of how our approach performs on finding adversarial shifts for a particular (synthetic) training distribution of attributes and faces.  However, we do not claim that the worst-case shifts here correspond to a particular real-world change in distribution, e.g., a distribution one would observe if a new set of CelebA images were collected.

---

### Official Review · Reviewer_Qr2b · 2022-07-11

**Rating:** 7
**Confidence:** 3
**Soundness:** 3 good
**Presentation:** 4 excellent
**Contribution:** 3 good

**Summary:**

In this work, the present a method to evaluate the robustness of models. This is done by first parametrizing the space of distribution shifts and then maximizing an approximation of the model loss over the shift parameter space. The approximation is done via second-order Taylor expansion and some theory is provided about the approximation error. Finally, they show in experiments the ability of their method to find interpretable distribution shifts that negatively affect a model.

**Questions:**

1. The supremum factor in Theorem 2 is hard for me to reason about. Are there reasonable settings where this bound can be simplified (e.g. Lipschitz losses)?
2. In Line 216, it is said that the variance of the Taylor approximation is $O(||\delta||^4)$, but shouldn't the second term in (8) contribute $O(||\delta||^2)$ variance? Should I be thinking of $\delta$ as bounded away from 0?
3. In section 3.3, the worst-case approximate loss is computed in a 2-norm ball. However, it seems to make more sense that the ball should be the sup-norm, with the idea being that each mechanism can be perturbed independently. Is it worth seeing what happens when you use that, or do you lose tractability?


**Limitations:**

1. I think the accuracy of the approximation needs to be better guaranteed in practice. It seems unclear based on just Thm 2 that the second-order Taylor will be accurate enough at moderately sized delta. One could perhaps verify the bound by sampling at the perturbed delta and checking the model's performance there, but that may not be always feasible.

**Strengths And Weaknesses:**

Strengths:
* Parameterizing shift space to probe biases is an original (to my knowledge) idea that could be a useful way to find interpretable biases.
* Good experiments
* Well written

Weaknesses:
* The approximation method could be too inaccurate in practice (see limitation 1)

---

> ### Author Response · Authors · 2022-08-02
> **Response**
>
> Thank you for the questions. We hope that the following clarifications are helpful.
>
> **The approximation method could be too inaccurate in practice (see limitation 1)...I think the accuracy of the approximation needs to be better guaranteed in practice. It seems unclear based on just Thm 2 that the second-order Taylor will be accurate enough at moderately sized delta. One could perhaps verify the bound by sampling at the perturbed delta and checking the model's performance there, but that may not be always feasible.**
>
> Accurate extrapolation to unseen distributions is a difficult task, and while there may be deltas for which our method is not accurate, alternative approaches may perform no better in predicting the loss. In situations where the variance of importance sampling is not too large, one can use our method to search for the worst case direction of shift, and then use importance weights to estimate the loss at the identified shift.
>
> For example, our updated experiments in Section 4.2 (see our general comment) indeed suggest that a plausible approach is to use the Taylor approach to select $\delta$, and then use the importance sampling estimator to evaluate the chosen $\delta$.
>
> **The supremum factor in Theorem 2 is hard for me to reason about. Are there reasonable settings where this bound can be simplified (e.g. Lipschitz losses)?**
>
> Theorem 2 is in a general form, which covers many different scenarios, where the bias could be very different (for example the amount of bias depends on the loss function and which variables are affected).
>
> We provide some intuition via an example (in Appendix C.3) where the approximation is exact, which occurs in linear-Gaussian models when using a squared error.
>
> We have added an additional example as Appendix C.8 in the revision, that illustrates how the bound can be simplified in a particular “covariate shift” setting, where $Y = f_0(X) + \epsilon$ and the distribution of $X$ is Gaussian, with a shift in the mean.  In this case, as one would expect, the approximation error depends on the error $f_0(x) - f(x)$, where $f(x)$ is our predictor, and different assumptions on how the modeling error $g(x) = f_0(x) - f(x)$ behaves over the domain.  In particular, the error in the approximation scales as the supremum (over $t \in [0, 1]$) of $\sqrt{var(g(X + t * \delta)^2 - g(X)^2)}$.  As a simple example, if $g(x) = f_0(x) - f(x)$ is a constant, i.e. if our predictor differs from the true function by an additive constant factor, the bound is $0$.
>
> **In Line 216, it is said that the variance of the Taylor approximation is O(|delta|^4), but shouldn't the second term in (8) contribute O(|delta|^2) variance? Should I be thinking of delta as bounded away from 0?**
>
> Apologies for the confusion: In this setting, we are interested in the behavior as $\delta$ becomes increasingly large, not small - hence, the $O(|\delta|^4)$ term dominates.  We have clarified this in the revision.
>
> **In section 3.3, the worst-case approximate loss is computed in a 2-norm ball. However, it seems to make more sense that the ball should be the sup-norm, with the idea being that each mechanism can be perturbed independently. Is it worth seeing what happens when you use that, or do you lose tractability?**
>
> The 2-norm ball has a few benefits:
> * First, in many cases we might expect the 2-norm ball to yield a worst-case shift that is easier for users to interpret: We might expect sup-norm constraints to result in maximum-size shifts in all conditionals, whereas the 2-norm constraint requires the algorithm to trade-off shifts in different conditionals, resulting in a ranking of importance. For example in Table 1 (left), we can identify that a shift in P(Bald | Female, Old) is more impactful than in other conditionals – a sup-norm constrained method may not yield such insight.
> * Second, there is a computational / optimization benefit, where the quadratic constraint results in a problem where strong duality holds, as discussed in the paper.
>
> That said, it is certainly possible to use other constraints (e.g., sup-norm), paired with a generic non-convex solver that allows for constrained optimization (similar to the one we use for optimizing the IS objective).  Here, tractability is problem-dependent:  While the IS approach (using a generic solver) is much slower on our problem than the Taylor approach, it still only takes ~2s to run.  We would expect that solving for sup-norm constraints would be feasible in the setting of Section 4.2.

---

> > ### Comment · Reviewer_Qr2b · 2022-08-07
> > **Response to authors**
> >
> > Thanks for the response! The responses had some good clarifications and the example provided in Appendix C is nice. I will maintain my score.

---

### Official Review · Reviewer_rEEg · 2022-07-13

**Rating:** 7
**Confidence:** 4
**Soundness:** 3 good
**Presentation:** 3 good
**Contribution:** 3 good

**Summary:**

The authors work out a method that, given a structural equation model having exponential family conditional densities, returns/computes the worst-case loss (i.e., negative log-likelihood) of any predictive model fitted to data arising from the SEM over a class of distributional perturbations to the SEM.  Computing the worst-case loss "naively" requires solving a (constrained) non-convex optimization problem, so the authors propose a vanilla sequential convex programming (trust region) method instead that seems to work well. The method is useful from an interpretability standpoint b/c it can tell us which conditionals (e.g., which "features") had to be wiggled the most in order to attain the worst-case loss.

**Questions:**

In addition to the questions embedded in the "Weaknesses" section above, here are a few more Q's for the authors:
* how would you do inference (e.g., get confidence intervals) for your estimation target (the worst-case loss)?
* it seems taking the worst-case over alternate graph **structures** (i.e., not just **parameters**) would be super useful -- any thoughts as to how tractable that might be?

**Limitations:**

Yes.

**Strengths And Weaknesses:**

Strengths:
* Very cool/timely topic, w/ an approach that is apparently well-thought through / totally reasonable
* The experiment in Sec 4.2 was neat
* Thm 1 is interesting/useful, though not surprising
* Clear / well-written paper

Weaknesses:
* I don't know that I totally get the approach here?  Why not define the worst-case loss to be taken over all distributions in a K-L (or any f-divergence, maybe) ball centered around the empirical measure?  I **think** the problem that falls out from that should be a (constrained) convex optimization problem.  Case closed.  Having said that, a difference in opinion on approach doesn't necessarily mean a paper is "no good" ...

* The advantage of the K-L type approach I just mentioned is you don't have to make distributional assumptions, whereas in your case you do.  Actually you assume the user knows quite a bit about the problem at hand: the SEM, that there exist conditional densities that are exponential family, and the shift function(s) -- is all that realistic?  Moreover, the shift function has to be specified on the natural parameter scale (e.g., on the $\mu/\sigma$ scale, in the case of a Gaussian density) -- how interpretable is that / can people easily think in those terms?

* Another potential approach: just draw data from several candidate SEM's (perhaps obtained by doing a very coarse grid search over your $\delta$ parameter) + take the worst-case loss over this (finite) set of SEM's.  This is very roughly in line w/ what you do in Sec 4.2.  Clearly this Monte Carlo-type approach cannot be **too** good, but I'm wondering how much worse it actually is from your approach ...

* There could probably be a few more / more thorough experiments in the paper ...

---

> ### Author Response · Authors · 2022-08-02
> **Response (Part 3)**
>
> **there could probably be a few more / more thorough experiments in the paper …**
>
> We have updated our experiment in Section 4.2 with more details on the performance of our method and importance sampling, as described in the general comment.  Some additional experiments are also included in Appendix C.5 (Figure 8), where we employ our method in a simulated setting with both binary and continuous variables. In particular, Figure 8 shows a scenario where the shift in a binary variable is so sufficiently simple that importance sampling is able to correctly predict the impact, but completely fails to capture the mean shift in a Gaussian variable. While our method exhibits more bias than IS in the binary variable, it predicts the shifted loss much more accurately in the Gaussian distribution.
>
> **how would you do inference (e.g., get confidence intervals) for your estimation target (the worst-case loss)?**
>
> If we split the sample, find a candidate shift $\delta$ using one split, and seek confidence intervals on the loss under this particular $\delta$, then this is straightforward:  We would estimate the loss using a re-weighting based approach (on the held-out sample), for which standard asymptotic theory provides confidence intervals.
>
> **it seems taking the worst-case over alternate graph structures (i.e., not just parameters) would be super useful -- any thoughts as to how tractable that might be?**
>
> We agree that such sensitivity analysis may be very useful in many applications. We present results on this question in Appendix B.2, referenced on line 124 in the original submission (line 125 in the revision), where we consider shifts which add new edges to the graph, as long as we don’t introduce cycles. This is just as tractable as our original formulation, as it just involves adding additional variables to the shift functions.

---

> ### Author Response · Authors · 2022-08-02
> **Response (Part 2)**
>
> **…you assume the user knows quite a bit about the problem at hand: the SEM, that there exist conditional densities that are exponential family, and the shift function(s) -- is all that realistic?... Moreover, the shift function has to be specified on the natural parameter scale (e.g., on the mu/sigma scale, in the case of a Gaussian density) -- how interpretable is that / can people easily think in those terms?**
>
> First, a small clarification, in case this was not entirely clear:  Our approach does not require full knowledge of the data-generating process, as discussed on lines 119-125 in the revision.  For instance, to model a change in the causal mechanisms of ordering $(O)$ laboratory testing, we only need to know (a) the causal parents of the variable $O$, (b) that this distribution is exponential family (which is true by definition for a binary / categorical RV), and (c) some user-supplied specification of the shift function.  If the user does not want to make strong assumptions about the shift function, they can specify a very flexible function at the expense of some interpretability (e.g., a neural network parameterized by $\delta$).  We do NOT need to know anything else about the causal graph or the other conditional probability distributions, nor do we need to estimate any of the other factors in the distribution other than $P(O \mid A, Y)$. In the revision, we have added some clarification on this point under Assumption 1.
>
> Second, the restriction to exponential families makes it easier to describe shifts in an interpretable fashion, even though they represent a non-trivial restriction for continuous variables.  To take the Gaussian example, we posit that it is easier to describe a change in the conditional mean vs. a fully general change in a conditional distribution, and easier to reason about whether or not a given change is “too large”.
>
> For the case of a Gaussian, we do show (in Example B.2 of the supplement) how to directly parameterize (for conditional Gaussians) a change in the conditional mean $\mu(z)$ by taking $s(Z; \delta) = \delta / \sigma(z)$, such that the natural parameter after shift becomes $\frac{\mu(z) + \delta}{ \sigma(z)}$.
>
> **Another potential approach: just draw data from several candidate SEM's (perhaps obtained by doing a very coarse grid search over your delta parameter) + take the worst-case loss over this (finite) set of SEM's. This is very roughly in line w/ what you do in Sec 4.2. Clearly this Monte Carlo-type approach cannot be too good, but I'm wondering how much worse it actually is from your approach**
>
> If we understand this idea correctly, it is to “simulate” from a new SEM under a particular choice of $\delta$, repeat several times, and take the worst shift.
>
> This approach would require far more information about the distribution (compared to our approach), to be able to simulate from the full joint distribution of $P_{\delta}$.  For instance, in Section 4.2, our method does not require any knowledge of $P(\text{Image} \mid \text{Attributes})$, i.e., our method does not require any ability to simulate new faces for a fixed set of attributes.  This is useful for scenarios where we only have a small number (e.g., 1000) of samples with a full set of attributes, where training e.g., a conditional GAN would be implausible.
>
> A very rough sense for how such a Monte-Carlo method would compare (if it had access to the true generative model) is given in Figure 5 (left), where the median shift found by our approach has better performance than 400 randomly chosen shifts.

---

> ### Author Response · Authors · 2022-08-02
> **Response (Part 1)**
>
> Thank you for your thoughtful review.  We address each point in turn, below.
>
> **Why not define the worst-case loss to be taken over all distributions in a K-L (or any f-divergence, maybe) ball centered around the empirical measure?  I think the problem should be a (constrained) convex optimization problem…The advantage of the K-L type approach I just mentioned is you don't have to make distributional assumptions, whereas in your case you do.**
>
> You are correct that the worst-case loss is much easier to derive for an f-divergence ball around the entire distribution.  Several coherent risk measures can be written in this way - a plug-in estimate of the Entropic Value-at-Risk (EVaR) with confidence level $1 - \alpha$ is precisely what you describe, a worst-case loss over all distributions $Q$ which are within a certain KL distance ($KL(Q \| \hat{P}) \leq - \ln \alpha$) of the empirical measure $\hat{P}$ [1].   Other examples include the Conditional Value-at-Risk (CVaR), which can be seen as an uncertainty set arising from a limiting f-divergence (see Example 3 of [2]).
>
> However, this simplicity comes with drawbacks.  In particular, these worst-case losses are very conservative, allowing all aspects of the distribution to change (e.g., both $P(X)$ and $P(Y \mid X)$), and do not provide any interpretable description of the resulting distribution.  CVaR, for instance, corresponds to simply sorting the training examples by loss, and taking the average of the top $\alpha$ fraction.
>
> To illustrate, consider CVaR and EVaR under the 0-1 loss for a classifier with 80% accuracy:  In both cases, for any $\alpha < 0.2$, they both yield a worst-case loss of $1.0$. For EVaR, this follows from the fact that the original loss is a binary random variable with $p = 0.2$, and that the Bernoulli distribution with $q = 1$ is within a KL-divergence ball of radius $(- \ln 0.2)$.  For CVaR, this follows from the fact that the worst-case 20% of the distribution contains all misclassified examples.
>
> Partially due to this overly-conservative behavior, there has been a line of work incorporating additional restrictions on the allowable shift (i.e., adding more assumptions).  For instance, [3] considers a worst-case shift similar to CVaR (a “worst-case subpopulation shift”), but where only $P(X)$ is allowed to change, and $P(Y \mid X)$ is assumed to be constant. Building on this line of work, [4] allows for changes in a conditional distribution, while retaining the same notion of a worst-case subpopulation.  Note that we compare our approach to [4] in Appendix D of the original submission.
>
> Our paper fits within this line of work, where we make additional assumptions on the nature of the shift, in order to construct sets of possible distributions where the worst-case loss is not overly conservative.
>
> Due to space limitations in the revision, we have included this discussion (and citations below) in Appendix B.4, but will move this into the main paper (if accepted) for the camera-ready version, which allows an additional page.
>
> [1] Ahmadi-Javid, A. (2012). Entropic Value-at-Risk: A New Coherent Risk Measure. Journal of Optimization Theory and Applications, 155(3), 1105–1123.
>
> [2] Duchi, J. C., & Namkoong, H. (2021). Learning models with uniform performance via distributionally robust optimization. The Annals of Statistics, 49(3), 1378–1406.
>
> [3] Duchi, J., Hashimoto, T., & Namkoong, H. (2020). Distributionally robust losses for latent covariate mixtures. arXiv Preprint arXiv:2007.13982.
>
> [4] Subbaswamy, A., Adams, R., & Saria, S. (2020). Evaluating Model Robustness and Stability to Dataset Shift. Proceedings of the 24th International Conference on Artificial Intelligence and Statistics.

---

> > ### Comment · Reviewer_rEEg · 2022-08-08
> > **Responses to the responses**
> >
> > Thanks for the responses.
> >
> > Responses to the responses:
> >
> > * Re: "In particular, these worst-case losses are very conservative, allowing all aspects of the distribution to change ..." -- wouldn't that be more realistic, though?  In practice, we never know whether the marginals or conditionals change.  So I'm not 100% sure I see this as a weakness (though I agree these approaches can certainly be conservative).
> >
> > * Re: "... we only need to know (a) the causal parents ..." -- well, ok sure, but what happens if you mis-specify the causal parents?
> >
> > * Re: "We present results on this question in Appendix B.2 ... where we consider shifts which add new edges to the graph ..." -- ok, thanks.  That's pretty neat.  I'll increase my score by 1.

---

> > > ### Author Response · Authors · 2022-08-09
> > > **Response**
> > >
> > > Thank you for taking the time to engage in discussion!  Some clarifications below, which we hope are helpful.
> > >
> > > ### Re: "very conservative", etc
> > >
> > > All we mean to say is that the right "worst-case" notion depends on the application and prior knowledge.  In some cases, conservative approaches (that allow the whole joint distribution to change) may be appropriate.  In other cases, where some prior knowledge exists, our approach provides a way to harness that knowledge.
> > >
> > > In a medical setting, for instance, the specificity/sensitivity of a diagnostic test is likely stable across hospitals, insofar as it depends only on the underlying biology.  In our lab testing example, this would correspond to assuming $P(L | Y, O)$ remains fixed, while approaches like CVaR/EVaR implicitly allow for this to change.
> > >
> > > ### Re: "what happens if you mis-specify the causal parents?"
> > >
> > > An additional clarification may be warranted here: In my response I said "to model a **change in causal mechanisms**...we only need to know (a) the causal parents..." (emphasis added).  In other words, **if we want to interpret the shift as causal** then we need to capture the causal parents.  Here, a change in $P(O | Y, A)$ has a particular causal interpretation as a change in laboratory testing policy.
> > >
> > > However, if we specify a different conditioning set for $O$ in the laboratory testing example (instead of $(A, Y)$), we will simply model different shifts in distribution, that may lack such a causal interpretation. Assumption 1, for instance, makes no mention of causality, just a set of changing and unchanging factors in the joint distribution, which we discuss briefly in Remark 1 (line 151).
> > >
> > > Consider what happens if we take a different conditioning set for $O$: We can always factorize the joint distribution over age $(A)$, disease $(Y)$, lab test order $(O)$, and lab test result $(L)$ in multiple ways to satisfy Assumption 1:
> > > * If we choose an empty conditioning set for $O$, we might factorize as $P(A, Y, O, L) = P(Y, A, L | O) P(O)$, where only $P(O)$ is changing, while $P(Y, A, L | O)$ is fixed.
> > > * If we choose only $Y$ as the conditioning set for $O$, we might factorize as $P(A, Y, O, L) = P(A, L | O, Y) P(O | Y) P(Y)$, where only $P(O | Y)$ is changing, and the other two factors are fixed.
> > >
> > > That is, we can specify a different set of variables to condition on, and we will still estimate a worst-case loss, but that worst-case loss will be over a *different set of distributions* than if we modeled a change in $P(O | A, Y)$.
> > >
> > > Of course, if we model a shift in $P(O)$, and the "true" shift occurs in $P(O | Y, A)$, then our approach may overstate/understate the worst-case loss (since the worst-case over shifts in $P(O)$ may not equal the worst-case over shifts in $P(O | Y, A)$).  But this is true of any method that assumes certain factors change, while other factors do not (e.g., those that assume $P(X)$ changes, but $P(Y | X)$ is fixed, or vice versa).

---

### Author Response · Authors · 2022-08-02
**Revised Version Uploaded**

Dear reviewers,

Thank you all for your thoughtful feedback - We have uploaded a revised version of the main paper and supplement, which incorporates clarifications discussed in our responses below.  These changes include (roughly in the order they appear):
* In response to comments by reviewer 93gd on the writing, we have revised the language to avoid vague references to “these” and “them” in the abstract and elsewhere, and clarified the distinction b/w Definitions 2 & 3.
* In response to comments by reviewer mdu8, we have revised Figure 1 to remove references to shifts in age, which proved confusing as we do not discuss such shifts until much later on in the paper.
* In response to reviewer mdu8, we have clarified some elements of the notation in the notation section and in Assumption 1.
* In response to reviewer Qr2b, we have clarified some elements of the discussion below Theorem 2.
* In response to comments by reviewers rEEg and 93gd regarding the depth of experiments, we include more aggregate statistics in Section 4.2 comparing the Taylor and IS methods across all simulation runs, and display the $\delta$ yielding the median drop in accuracy in Table 1 (left), as a more representative shift than the randomly chosen $\delta$ we displayed previously.
* Throughout, we have fixed some minor typos.

Since the revision is still limited to 9 pages, we have placed additional content into appendix sections, but **if the paper is accepted, we will incorporate these additions into the camera-ready (which allows for 10 pages)**.  This content includes
* In response to questions by reviewer 93gd (about the relationship to approaches like Mandoline, GDE, and ATC) and a question by reviewer rEEg (about using simple f-divergence robustness sets), we have added Appendix B.4 “Comparisons to other approaches”, which codifies our response to those points (see our responses to those reviewers for more detail).
* In response to questions by reviewer Qr2b and mdu8 about the accuracy of the second-order approximation, we have added an example (as Appendix C.8) where this bound can be simplified further, as discussed in our responses to those reviewers.
* In response to a concern by reviewer 93gd that the shift in Section 4.2 only yields a small drop in accuracy, we include additional experimental results (as Appendix E.4) where we demonstrate much larger drops in accuracy for larger values of the constraint $\lambda$ (e.g., an average drop of 23% for $\lambda = 8$).

Finally, we have re-run the experiment in Section 4.2 after discovering a mistake in the code that impacted the Importance Sampling (IS) estimator, which is now fixed:  Previously, the IS estimator was calculating the importance weights using an incorrect probability distribution.  Recall that each estimator can be used in two ways: To **find** a worst-case shift (by optimizing over the estimate), and to **evaluate** a given shift (by estimating the loss under that shift).  After re-running the experiment, we observe the following:
* Using IS to evaluate the shift found by the Taylor approach now gives comparable results to using the Taylor approximation to evaluate the same shift.  In Table 1 (right) of the previous version, the IS estimate (for a single shift) was 0.792 and Taylor estimate was 0.878 with a ground-truth of 0.887.  On lines 300-301 in the revision, we now report their RMSE is comparable (1.91% vs. 1.92%) and in Table 1 (right) we observe their average estimates are nearly identical, when evaluating the shift found by the Taylor approach.
* However, the Taylor approach now finds a more adversarial shift in 96% of cases (caption of Figure 5), which is even better than the 73% previously reported; on further analysis, we believe this is because the coding error resulted in an implicit bias that – for this specific synthetic distribution – inadvertently had helped IS find more impactful shifts.
* When IS is used for **both** finding a shift, and evaluating the same shift, the mean absolute error (MAE) is 6.9% (line 302 in revision, previously 17.6% on line 310 in original version), which can be compared to 1.5% MAE when using the Taylor approach for both finding and evaluating a shift (line 303 in the revision). This may suggest that optimizing the IS objective is prone to “overfitting”, choosing a sub-optimal $\delta$ from a high-variance region.

---

### Meta-Review · Area_Chair_6v4N · 2022-08-21

**Recommendation:** Accept
**Confidence:** Certain

**Metareview:**

This paper studies the important problem of structured distribution shifts. Departing from previous works that use worst-case regions based on distance measures (f-divergences, Wasserstein distances), the authors construct an interesting framework based on a (partially) specified structural causal model. The proposed framework assumes a number of problem-specific structures, which I view as a major advantage over previous approaches. However, the methodology is interpretable insofar as the causal graph is partially specified and the shift function is structured. For the proposed approach to have applied impact, I recommend identifying realistic application scenarios where these structures are prominent, going beyond toy examples.

Furthermore, please contextualize the work with respect to the following related papers:

Taylor expansions for small robustness radius: Lam, Robust sensitivity analysis for stochastic systems, Mathematics of Operations Research, 2016.

Structured distribution shift based on SEMs: Heinze-Deml, Meinshausen, Conditional Variance Penalties and Domain Shift Robustness, Machine Learning, 2021.

Robustness evaluation: Li, Namkoong, Xia, Evaluating model performance under worst-case subpopulations, NeurIPS, 2021.

**Award:**

No

---

### Decision · Program_Chairs · 2022-09-14

Accept